# Denoising and Motion Artifact Removal Using Deformable Kernel Prediction Neural Network for Color-Intensified CMOS

**DOI:** 10.3390/s21113891

**Published:** 2021-06-04

**Authors:** Zhenghao Han, Li Li, Weiqi Jin, Xia Wang, Gangcheng Jiao, Xuan Liu, Hailin Wang

**Affiliations:** 1Key Laboratory of Photo-Electronic Imaging Technology and Systems, School of Optics and Photonics, Ministry of Education of China, Beijing Institute of Technology, Beijing 100081, China; 3120160303@bit.edu.cn (Z.H.); jinwq@bit.edu.cn (W.J.); wx_may@263.net (X.W.); 3120205295@bit.edu.cn (X.L.); 3120180569@bit.edu.cn (H.W.); 2Science and Technology on Low-Light-Level Night Vision Laboratory, Xi’an 710065, China; jiaogc613@163.com

**Keywords:** color-intensified CMOS, liquid-crystal tunable filter, image denoising, motion artifacts, convolutional neural network

## Abstract

Image intensifiers are used internationally as advanced military night-vision devices. They have better imaging performance in low-light-level conditions than CMOS/CCD. The intensified CMOS (ICMOS) was developed to satisfy the digital demand of image intensifiers. In order to make the ICMOS capable of color imaging in low-light-level conditions, a liquid-crystal tunable filter based color imaging ICMOS was developed. Due to the time-division color imaging scheme, motion artifacts may be introduced when a moving target is in the scene. To solve this problem, a deformable kernel prediction neural network (DKPNN) is proposed for joint denoising and motion artifact removal, and a data generation method which generates images with color-channel motion artifacts is also proposed to train the DKPNN. The results show that, compared with other denoising methods, the proposed DKPNN performed better both on generated noisy data and on real noisy data. Therefore, the proposed DKPNN is more suitable for color ICMOS denoising and motion artifact removal. A new exploration was made for low-light-level color imaging schemes.

## 1. Introduction

Since the color level that a human eye can distinguish is hundreds of times greater than the grayscale level [1], color imaging can make full use of the sensitivity of the human eye to color information, thereby improving its ability of scene understanding. Therefore, color imaging in low-light-level conditions has outstanding advantages in the areas of the military, security, etc., compared with traditional grayscale imaging, and it shows a wide range of application prospects. Low-light-level imaging techniques which use an image intensifier as the core imaging device have been used in advanced military night-vision for many years. An image intensifier can still imaging under an environmental illuminant level of 10^−4^ lx. A typical grayscale image of an image intensifier is shown in Figure 1a. The image intensifier is a direct-view imaging device, the phosphor screen of which is directly observed by the human eye. The image intensifier has a higher sensitivity than a CMOS/CCD. However, a CMOS/CCD has digital outputs which can be conveniently processed and stored. In order to combine their advantages, an intensified CMOS/CCD (ICMOS/ICCD) was proposed by coupling the phosphor screen of the image intensifier to the CMOS/CCD sensor. Generally, the ICMOS/ICCD has better imaging performance compared with low-light-level imaging devices such as sCMOS and EMCCD. With the development of the sensitivity of third-generation image intensifiers, it is of great significance to expand their application to low-light-level color imaging through the modulation of filtering elements such as a filter wheel [2,3,4] and a liquid-crystal tunable filter (LCTF) [5,6], whereby it may be possible for low-light-level color imaging techniques to expand their extreme working illuminant level to 10^−3^ lx or even 10^−4^ lx.

The performance of a low-light-level imaging device is not only related to the characteristics of the imaging device itself, but also the image denoising algorithm. Image denoising algorithms based on convolutional neural networks (CNNs) have achieved good denoising results in recent years. Unlike the imaging scheme of a CMOS/CCD, which simultaneously obtains various color information at different spatial pixel locations through a color filter array (CFA), due to the separation of the photosensitive surface and the digital sampling surface, the ICMOS/ICCD has to obtain various visible-band color information through filtering elements in a time-division manner. Although obtaining color information in a time-division manner can obtain full-resolution color images without the need to design additional demosaicing algorithms, if there is a motion target in the scene or if image device shaking exists, color-channel motion artifacts would be introduced, as shown in Figure 1b. Therefore, when designing denoising algorithms for time-division color imaging devices, it is necessary to take the motion artifacts into consideration.

Research on low-light-level color image denoising has explored a wide variety of technical routes. Owing to the development of neural network techniques, CNN-based low-light-level color image denoising methods with satisfactory denoising results have been proposed in recent years. However, commonly used color image denoising algorithms are mainly designed for single-frame denoising and, thus, do not consider motion artifacts. Although multi-frame color image denoising algorithms need to consider the matching of moving targets between adjacent frames, they are all aimed at the CFA-based color imaging scheme, which is different from the motion artifacts in a time-division color imaging scheme in terms of image characteristics and correction strategies; thus, they cannot be directly applied to ICMOS/ICCD denoising. Considering the relevance to the research in this article, we mainly introduce two types of methods: spatial-filter-based denoising and CNN-based denoising.

(1)Spatial-filter-based denoising methods

Buades et al. [7] proposed a nonlocal means (NLM) denoising method, which selects pixels with similar structures in the image to perform a weighted average filtering operation, circumventing the limitations of the local information of images. Dabov et al. [8,9] proposed the BM3D denoising method, which used block matching and aggregation strategies to obtain three-dimensional image blocks, before performing denoising on the basis of collaborative Wiener filtering. BM3D remains one of the benchmark algorithms in the area of image denoising. Knaus et al. [10] proposed a progressive image denoising method that iteratively processes the original noisy image and the denoised image to obtain a better result.

(2)CNN-based denoising methods

Mao et al. [11] proposed the denoising network RED with skip connections, inspired by the auto-encoder network UNet proposed by Ronneberger et al., for medical image segmentation [12]. Zhang et al. [13] proposed the denoising network DnCNN, the structure of which was based on the VGG [14] network, whereas batch normalization and a residual learning strategy were also introduced. Tai [15] proposed the denoising network MemNet, the long-term memory of which was set up through the skip connections between memory blocks. Liu et al. [16] proposed the denoising network WIN, which learned the noise prior of noisy images. Lefkimmiatis et al. proposed the denoising networks NLCNN [17] and UDNet [18], which embedded nonlocal operation units into the network. Kligvasser et al. [19] proposed the basic block of a denoising network named xUnit, which was trained with fewer parameters by replacing the convolutional units with xUnits. Ulyanov et al. [20] performed image denoising through network structure learning instead of weight learning. Cha et al. [21] proposed the denoising network FC-AIDE, the parameters of which were fine-tuned by combining a regularization method and learned image priors. Liu et al. [22] proposed the denoising network MWCNN, which replaced the pooling and up-convolution operations with wavelet operations. Mildenhall et al. [23] proposed the kernel prediction network (KPN) for multi-frame denoising. Image misalignment was added to training data to simulate camera motion, such that small misalignment could be corrected during denoising. Plotz et al. [24] proposed the differentiable relaxation KNN and nonlocal denoising network N3Net. Liu et al. [25] proposed the nonlocal denoising network NLRN, which achieved end-to-end training of a nonlocal network. Zhang et al. [26] proposed FFDNet, which could handle a wide range of noise levels and spatial variation noise. Lehtinen et al. [27] trained denoising networks using noisy images without using noise-free images. Liu et al. [28] proposed the BayerUnify network for raw color image denoising instead of sRGB color image denoising. Wang et al. [29] proposed the EDVR network for video denoising, which introduced submodules with different functions and adopted a residual learning strategy. Guo et al. [30] proposed the blind denoising network CBDNet, which introduced a noise level estimation subnet. Marinc et al. [31] proposed multi-KPN (MKPN), which trained multiple filter kernels with different spatial sizes and fused them when performing inference, thereby increasing the calculation efficiency of the MKPN. However, this network can still only handle small misalignment, as with KPN. Xu et al. [32,33] proposed the STPAN, which can be used for single-frame denoising and video denoising. STPAN learned deformable kernels instead of fixed-shape kernels, while it could be applied to solve large misalignment of adjacent frames in motion video scenes. Tassano et al. [34] proposed the DVDNet for video denoising, which introduced optical flow estimation for temporal denoising and relied heavily on the accuracy of optical flow estimation. The authors later improved the network and proposed the FastDVDNet [35], replacing the explicit optical flow estimation with a learning process. Tim et al. [36] proposed a training data generation method using an unprocessing approach for denoising. Ronnachal et al. [37] proposed a data generation method based on an image processing pipeline.

Denoising networks based on kernel prediction strategies such as KPN, MKPN, and STPAN have the ability to simultaneously perform image denoising and image registration. KPN and MKPN can only handle small misalignment of less than 16 pixels. Since STPAN adopts deformable kernel prediction, it has the potential to solve large misalignment between frames; however, it is mainly aimed at simultaneous color imaging schemes based on CFA. Due to the simultaneous imaging of each color channel, motion artifacts only exist between adjacent frames instead of color channels of a single frame. STPAN requires a burst of images as input and output of a denoised single-frame-prediction through sampling pixels among input images. For color ICMOS images with motion artifacts between adjacent color channels, STPAN can only treat the input single-frame color image as continuous three-frame grayscale images and output a single-frame grayscale prediction. Hence, it cannot be directly used to solve motion artifacts between adjacent color channels caused by time-division imaging. Therefore, there is currently no denoising algorithm suitable for LCTF-ICMOS images with color-channel motion artifacts. Considering that STPAN can solve large misalignment, we propose a denoising network named DKPNN that can adapt to color-channel motion artifacts using STPAN as inspiration.

A deformable kernel prediction neural network (DKPNN) is proposed for joint denoising and motion artifact removal in this article. A data generation strategy for color-channel motion artifacts is also presented, and the proposed DKPNN was trained using the generated data. This network can simultaneously complete denoising and motion artifact removal. We set up an LCTF-ICMOS experimental color imaging system to verify the effectiveness of the DKPNN. The innovations of this article are as follows:(1)A data generation strategy for color-channel motion artifacts;(2)A DKPNN which achieves joint denoising and motion artifact removal, suitable for color ICMOS/ICCD;(3)An LCTF-ICMOS low-light-level color imaging system with high integration.

The remainder of this article is arranged as follows: Section 2 details each step of the proposed DKPNN for joint denoising and motion artifact removal; Section 3 describes the setup of the LCTF-ICMOS experimental imaging system; Section 4 presents the validation results of the proposed method and discusses its effectiveness; and Section 5 concludes the paper and discusses future works.

## 2. Joint Denoising and Motion Artifact Removal Using DKPNN for Color LCTF-ICMOS

The pipeline and network structure of the proposed DKPNN are shown in Figure 2. The proposed DKPNN contains two parts: an offset-prediction subnet and a weight-prediction subnet. The offset-prediction subnet predicts the position of sampling pixels in each channel of the input color image, and the weight-prediction subnet predicts the weights of the resampled pixels. A noisy image with color-channel motion artifacts is first input to the offset-prediction subnet with a UNet structure. The offset-prediction network simultaneously outputs the predicted sampling positions of the three-dimensional deformable kernel in the spatiotemporal domain and the extracted features of the input color image. The resampled pixels are then calculated after applying trilinear interpolation to the predicted sampling positions. Next, the resampled pixels, the input image, and the extracted features are concatenated and are input to the weight-prediction subnet, which is composed of three convolutional layers. The weight-prediction subnet then outputs predicted weights. Lastly, the denoised image with motion artifacts removed is calculated through weighted averaging of the resampled pixels with the predicted weights.

### 2.1. Joint Denoising and Motion Artifact Removal Using DKPNN

#### 2.1.1. Offset-Prediction Subnet

The essence of the image denoising algorithm is to utilize the zero-mean statistical characteristic of image noise. An estimate of a noise-free image can be obtained by averaging multiple irrelevant samplings of the same image signal. Hence, the effectiveness of denoising comes from the averaging operation in a statistical sense. A general image model containing noise can be expressed as
(1)I(x,y)=g(x,y)+σ(g(x,y))ξ(x,y),
where *I*(*x*,*y*) denotes the noisy image signal observed by the imaging system, *x* and *y* denote the spatial location of a pixel, *g*(*x*,*y*) denotes the unknown noise-free ground truth image signal, whereby the goal of the denoising algorithm is to estimate *g*(*x*,*y*), ξ(*x*,*y*) denotes the signal-independent zero-mean random noise with a standard deviation of 1, and σ denotes the overall noise distribution of the image as a function of *g*(*x*,*y*).

If the adopted noise model is signal-independent additive white Gaussian noise (AWGN), then the image noise is a zero-mean normal distribution with variance $\sigma_{0}{}^{2}$, as expressed below.
(2)$$\sigma (g(x,y))=\sigma_{0},$$
(3)$$\sigma_{0}\cdot \xi (x,y)∼N(0,{\sigma_{0}}^{2}),$$
where N denotes a normal distribution.

If the adopted noise model is signal-dependent Poisson–Gaussian noise (PGN), then it is expressed as
(4)$$\sigma (g(x,y))\cdot \xi (x,y)∼P(\lambda )\approx N(0,\sigma_{p}\cdot g(x,y)+{\sigma_{g}}^{2}),$$
where P denotes a Poisson distribution, which is generally estimated by Gaussian distribution as shown in Equation (4) [38,39,40], while $\sigma_{p}$ and $\sigma_{g}$ denote the signal-dependent noise level and signal-independent noise level, respectively. In Equation (1), if the expectation of ξ(x,y), E(ξ(x,y))=0, then
(5)E(I(x,y))=g(x,y).

In traditional temporal averaging denoising algorithms, the estimation of the noise-free image is e(x,y)=E(I(x,y)). However, this has obvious limitations. First, the image sequence must be input instead of a single-frame image. Secondly, if there are moving objects in the scene or if the device is shaking, motion artifacts may be introduced. In spatial filter-based single-frame image denoising, the estimation of the noise-free image satisfies
(6)$$e(x,y)=\sum_{i=1}^{n}I(x+\Delta x_{i},y+\Delta y_{i})\cdot w(x,y,i),$$
where $\Delta x_{i}$ and $\Delta y_{i}$ denote the offsets of pixel (*x*,*y*), which represents the sampling process of spatial pixels, i.e., the selection strategy of pixel positions which would be involved in the weighted average operation. In typical two-dimension rectangular filter kernels, ($\Delta x_{i}$,$\Delta y_{i}$) is the eight-neighbor offset of target pixel (*x*,*y*), i.e., $\Delta x_{i}\in \{-1,0,1\}$ and $\Delta y_{i}\in \{-1,0,1\}$. The filtering process of a typical rectangular filter kernel is shown in Figure 3a, where *n* denotes the total number of pixels in the filter kernel, and *w* (*x*,*y*,*i*) denotes the corresponding weight of the filter kernel.

As shown in Equation (6), the key to the spatial filtering denoising operation lies in the design of the pixel sampling strategy and corresponding weights. Spatial filter-based denoising methods design *w* (*x*,*y*,*i*) artificially, whereas CNN-based denoising methods learn *w* (*x*,*y*,*i*) in a data-driven manner and achieve better results when using a large number of convolution kernels. Generally, the filter kernels learned by a CNN have fixed parameters and are image content independent. The same filtering operation is used for all image scenes; thus, it is not flexible. Therefore, we adopted the deformable kernel prediction strategy (as shown in Figure 3b) proposed in [33], which simultaneously predicts the spatial pixel sampling location and the corresponding weights. Instead of directly predicting ($\Delta x_{i}$, $\Delta y_{i}$) in Equation (6), the sub-pixel accuracy offsets relative to a typical rectangular filter kernel were predicted, which further reflects the idea of deformation, as expressed below.
(7)$$e(x,y)=\sum_{i=1}^{n}I(x+\Delta x_{i}+\Delta u_{i},y+\Delta y_{i}+\Delta v_{i})\cdot w(x,y,i).$$

If the offset between adjacent color channels is also taken into consideration and if the deformable kernel prediction is extended to color channels, then
(8)$$e(x,y,c)=\sum_{i=1}^{n}I(x+\Delta x_{i},y+\Delta y_{i},c+\Delta c_{i})\cdot w(x,y,c,i),$$
where *c* denotes the index of the color channel, i.e., c∈{0,1,2}.

We adopted the same strategy as [33], whereby, instead of directly predicting ($\Delta x_{i}$, $\Delta y_{i}$, $\Delta c_{i}$), the offsets ($\Delta u_{i}$, $\Delta v_{i}$, $\Delta k_{i}$) relative to a three-dimensional cubic filter kernel were predicted, i.e.,
(9)$$e(x,y,c)=\sum_{i=1}^{n}I(x+\Delta x_{i}+\Delta u_{i},y+\Delta y_{i}+\Delta v_{i},c+\Delta c_{i}+\Delta k_{i})\cdot w(x,y,c,i).$$

The learning procedure of three-dimension deformable sampling locations is shown in Figure 4. A size of 3 × 3 × 3 was adopted as the kernel size of the three-dimensional deformable kernel for each color channel, i.e., *n* = 27.

After the prediction of the offsets, the input image is resampled using a trilinear interpolation method according to the predicted offsets, as expressed below.
(10)$$I(x+\Delta x_{i}+\Delta u_{i},y+\Delta y_{i}+\Delta v_{i},c+\Delta c_{i}+\Delta k_{i})=I(p,q,j)=\sum_{i=0}^{7}s_{i}I_{i},$$
where *I_i_* denotes the nearest eight vertices of the position that needs to be resampled, while *s_i_* denotes the corresponding sampling coefficient. They satisfy
(11)$$\left\{ \begin{array}{l} I_{0}=I(x_{0},y_{0},c_{0})\\ I_{1}=I(x_{0},y_{0},c_{1})\\ I_{2}=I(x_{0},y_{1},c_{0})\\ I_{3}=I(x_{0},y_{1},c_{1})\\ I_{4}=I(x_{1},y_{0},c_{0})\\ I_{5}=I(x_{1},y_{0},c_{1})\\ I_{6}=I(x_{1},y_{1},c_{0})\\ I_{7}=I(x_{1},y_{1},c_{1}) \end{array}\right.\;\;\;\;\;\;\;\;\left\{ \begin{array}{l} s_{0}=(x_{1}-p)\cdot (y_{1}-q)\cdot (c_{1}-j)\\ s_{1}=(x_{1}-p)\cdot (y_{1}-q)\cdot (j-c_{0})\\ s_{2}=(x_{1}-p)\cdot (q-y_{0})\cdot (c_{1}-j)\\ s_{3}=(x_{1}-p)\cdot (q-y_{0})\cdot (j-c_{0})\\ s_{4}=(p-x_{0})\cdot (y_{1}-q)\cdot (c_{1}-j)\\ s_{5}=(p-x_{0})\cdot (y_{1}-q)\cdot (j-c_{0})\\ s_{6}=(p-x_{0})\cdot (q-y_{0})\cdot (c_{1}-j)\\ s_{7}=(p-x_{0})\cdot (q-y_{0})\cdot (j-c_{0}) \end{array}\right.,$$
where: *x*_0_ and *x*_1_ denote the pixel locations that are closest to the pixel location to be sampled along the *x*-direction; *y*_0_ and *y*_1_ denote the pixel locations that are closest to the pixel location to be sampled along the *y*-direction; and *c*_0_ and *c*_1_ denote the pixel locations that are closest to the pixel location to be sampled along the color-channel direction. The resampling process through the trilinear interpolation method is shown in Figure 5.

The offset-prediction subnet predicts offsets relative to a three-dimensional cubic filter kernel in each color channel, which is a typical pixel-wise prediction task. The UNet is an encoder-decoder network extracting multi-scale features, where the same-scale features between encoder and decoder are connected with skip connections. It performs well in a wide range of pixel-wise prediction tasks. Therefore, the offset-prediction subnet adopted an auto-encoder structure similar to UNet. Multi-scale image features are also extracted while performing offsets prediction. By inputting the extracted multi-scale image features along with the resampled pixels into the weight-prediction subnet, there is no need to re-extract image features when performing weight prediction, thereby reducing the depth of the weight-prediction subnet and reducing the amount of model parameters. Therefore, the extracted features were also made output by the offset-prediction subnet. To achieve this goal, the first 21 convolution layers of the offset-prediction subnet, namely L1–L21, adopt the same number of feature channels as UNet. The 256-channel feature of L21 is divided into two parts of 128-channel feature, which are respectively used as 128-channel image feature output and offsets output through L22–L24 and L25–L27. For a three-dimensional cubic filter kernel with *n* sampling pixels, offsets along three directions are predicted for each sampling pixel and the input image contains three color channels, thus, the number of feature channels of the final output by L27 is *n* × 3 × 3. The number of channels in each convolutional layer of the offset-prediction subnet is shown in Table 1.

#### 2.1.2. Weight-Prediction Subnet

Since the predicted weights are to be calculated with the resampled pixels, jointly inputting the resampled pixels along with input image and extracted image features to the weight-prediction subnet is beneficial for performing weight prediction. Therefore, the input image, resampled pixels, and image features were concatenated as concatenated features and were input to the weight-prediction subnet. Normal CNN-based denoising algorithms directly learn the mapping between noisy and noise-free image pairs, and the learned filter kernels are fixed and scene-independent. Unlike the normal training strategy, the weights predicted by the proposed weight-prediction subnet were referred to the input image scene and image features. The learned kernels were, thus, scene-dependent. Moreover, since the image features were extracted by the offset-prediction subnet, a shallow model is enough for weight prediction, which reduced the amount of model parameters. Finally, a structure containing three convolutional layers was adopted. The number of channels of L1 and L2 are both 64. Since the resampled pixels has *n* × 3 channels, the number of channels of L3 should also be *n* × 3. The number of channels in each convolutional layer is shown in Table 2. After the weighs were predicted, the final denoised image with motion artifacts removed could be calculated according to Equation (9).

### 2.2. Data Generation Method for Color-Channel Motion Artifacts

Since it is difficult to collect a large number of images containing color-channel motion artifacts, we used the open-access video dataset Adobe240 [41] with different kinds of motion objects to generate the needed data. The generated data were used to train the proposed DKPNN. The Adobe240 dataset contains two types of motion: object motion and device motion. It contains 133 video clips with a frame rate of 240 fps and a resolution of 1280 × 720. In order to simulate data with color-channel motion artifacts, we extracted the R, G, and B channels of every three adjacent frames of the original 240 fps video and combined them as the color channels of a single frame to generate an 80 fps video. In order to generate training data with color-channel motion artifacts of different severity, the original video frame rates were down sampled to 120 fps and 80 fps, respectively, and the same data generation strategy was adopted to obtain 40 fps and 20 fps videos. The generated videos with different frame rates were used to represent data describing small motion, medium motion, and big motion to train the network. The generated images with different motion severity are shown in Figure 6.

When generating training data, box filtering was first adopted to down sample the training image four times (two times each horizontally and vertically) to reduce the impact of other kinds of artifacts such as compression artifacts. Then, an inverse gamma correction, as used in [36], was adopted for both ground truth images and generated images to convert the image signal back into a linear response. The inverse gamma correction process can be expressed as
(12)$$\Gamma^{-1}(I)=max{\left(I,\varepsilon \right)}^{2.2},$$
where *I* denotes the input image, and ε denotes a constant factor preventing numerical instability during training; ε = 10^−8^ was adopted in this article.

After inverse gamma correction, image noise, described by the PGN model shown in Equation (4), was added to the image as the noisy input of the network, where $\sigma_{p}$ was randomly sampled from [10^−4^, 10^−2^] and $\sigma_{g}$ was randomly sampled from [10^−3^, 10^−1.5^].

### 2.3. Training

We adopted the same loss function as in [33], i.e., L1 loss with a grouped annealing term. The *n* × 3 sampling points were divided into *q* groups $\left\{\Phi_{0},\Phi_{1},\ldots ,\Phi_{q-1}\right\}$. At the early stage of training, the annealing term was the main loss. Each group of sampling points was stimulated to independently predict the pixel offset, the idea of which is similar to the Dropout [42] strategy. On one hand, the group strategy can prevent the training parameters from heavily relying on a few sampling points. On the other hand, due to the reduction in the number of sampling points, the grouped sampling points are stimulated by different color channels to obtain a better result of removing motion artifacts, as expressed below.
(13)$$e_{i}(x,y,c)=k\cdot \sum_{j\in \Phi_{i}}I(x+\Delta x_{j}+\Delta u_{j},y+\Delta y_{j}+\Delta v_{j},c+\Delta c_{j}+\Delta k_{j})\cdot w(x,y,c,j).$$

Then, the total loss function can be expressed as
(14)$$L(e,g)={\left\|\Gamma^{-1}(e)-\Gamma^{-1}(g)\right\|}_{1}+\eta \gamma^{t}\sum_{i=1}^{q}{\left\|\Gamma^{-1}(e_{i})-\Gamma^{-1}(g)\right\|}_{1},$$
where *e* denotes the predicted denoised image with motion artifacts removed, *g* denotes the ground truth image, ${\left\|\cdot \right\|}_{1}$ denotes L1 loss, η and γ are hyperparameters of the annealing term, and *t* denotes the current training step; η=100, γ=0.9998, and *q* = 3 were adopted in this article.

The relationship between the coefficient of the annealing term and the number of training steps is shown in Figure 7. At the early stage of training, the coefficient of the annealing term was much greater than 1, and the annealing loss was the main loss for optimization. As the number of training steps increased, the coefficient of the annealing term gradually decreased. After 23,000 training steps, the coefficient was about 1, exhibiting the same effect as L1 loss. After 35,000 training steps, the coefficient of the annealing term was much lower than 1; thus, L1 loss was the main loss for optimization.

The training data generated according to the method described in Section 2.2 were randomly cropped as 128 × 128 image blocks during training. Random horizontal flips and random vertical flips were adopted as a data augmentation strategy. The batch size was set to 16. The AdamW [43] optimizer was used to optimize the network parameters, and the weight decay was set to 0.9. The initial learning rate of the network was 1 × 10^−4^, while it was 1 × 10^−5^ after 40,000 steps. The relationship between the number of training steps and the learning rate is also shown in Figure 7. The network was trained for 50,000 steps on an NVIDIA GTX1080Ti, which took about 4.2 days.

## 3. Experimental Setup

The LCTF-ICMOS imaging system was improved on the basis of the results in [6]. The system was mainly composed of an objective lens, the LCTF and its controller, the GaAsP third-generation image intensifier, the relay lens, a low-light-level CMOS, and a PC. The imaging process of the LCTF-ICMOS is shown in Figure 8a. The radiant energy of the illuminant is reflected by the scene, and the reflected light is received by the objective lens of the system, which then passes through the LCTF and is focused on the photocathode of the image intensifier. Then, the phosphor screen outputs a monochrome image, which is captured by the CMOS. The LCTF sequentially changes the R, G, and B filter states driven by the synchronization signal, and the CMOS synchronizes capturing the phosphor screen of the image intensifier to obtain the corresponding R, G, and B images. After an RGB cycle, the captured adjacent R, G, and B frames are reconstructed as the final color output.

A PENTACON 50 mm fixed focus lens with a maximum F number of 1.8 was adopted as the objective lens. The tri-color LCTF of Meadowlark Optics was adopted. This LCTF contains a liquid-crystal module and a controller, which can switch R, G, and B filter states. The spectral transmittance functions of the R, G, and B filter states are shown in Figure 9a. The GaAsP third-generation image intensifier was adopted, the spectral sensitivity of which is shown in Figure 9b. The integral sensitivity of the image intensifier was 1525 μA/lm, the brightness gain was greater than 9000 cd/m^2^/lx, and the resolution was 54 lp/mm. The relative response function of the LCTF and image intensifier is shown in Figure 9c. A Canon EF-S 35 mm Is STM lens was adopted as the relay lens. The Photonis NOCTURN low-light-level monochrome CMOS was adopted with a resolution of 1280 × 1024, a pixel size of 9.7 μm, a maximum frame rate of 100 Hz, a dynamic range of more than 60 dB, and a readout noise of less than 4e^−^. The final LCTF-ICMOS imaging system setup is shown in Figure 8b.

## 4. Results and Discussion

### 4.1. Generated Noisy Data

Since the ground truth data could be obtained for the generated noisy data, full-reference image quality assessment (FRIQA) metrics were adopted to evaluate the denoising results of each algorithm, namely, the peak signal-to-noise-ratio (PSNR) [44] and structural similarity index measure (SSIM) [45].

In order to demonstrate the effectiveness of the proposed method, the results were compared with those obtained using CBM3D [8], DnCNN [13], fDnCCNN [13], and FFDNet [26]. Because these denoising algorithms were not designed for color-channel motion artifact removal, the dense optical flow estimation (FE) method proposed by Farnebäck et al. [46] was adopted after denoising of the compared algorithms for color-channel registration. Then, these images were compared with the results of the proposed DKPNN.

Noise was added to the test set images using the PGN model shown in Equation (4), and two sets of noise levels were tested, where $\sigma_{p}$ = 4 × 10^−3^, $\sigma_{g}$ = 3 × 10^−2^ and $\sigma_{p}$ = 2 × 10^−3^, $\sigma_{g}$ = 2 × 10^−2^, respectively. The results of each algorithm on the generated test set data with different noise levels are similar and the results on lower noise level are shown in Figure 10. Note that we manually blurred the human faces in Figure 10 for privacy protection. The PSNR and SSIM metrics for each algorithm on the test set data with different noise levels are shown in Table 3.

The denoising results of the algorithms compared on the test set data were generally acceptable. However, if the compared algorithms did not use FE for color-channel registration, they would have been unable to deal with the color-channel motion artifacts, as shown in Figure 10c. After FE was applied, severe blur was introduced after color-channel registration, and the image details were lost.

For the generated test set data from Scene 1, the compared algorithms performed well at the edges of houses with rich details, but introduced severe blur in the regions with the lawn, sky, and moving bicycle. By contrast, the proposed DKPNN could better correct the color-channel motion artifacts and maintain image details in all the above-mentioned regions.

For the generated test set data from Scene 2, fDnCNN, FFDNet, and the proposed DKPNN achieved good denoising results, whereas residual noise could be observed in the images processed by CBM3D and DnCNN. The compared algorithms after applying FE introduced severe blur or artifacts in the details of the trophy, the reflection of the trophy, and the edge of the iron box, whereas the proposed DKPNN could better maintain the image details.

For the generated test set data from Scene 3, the compared algorithms successfully performed denoising. After applying FE, blur or artifacts were introduced on the edges of the moving bicycle, house, and plants, whereas the proposed DKPNN better maintained the image details.

For the generated test set data from Scene 4, fDnCNN, FFDNet, and the proposed DKPNN performed well in denoising the billiard table; however, residual noise could be observed in the images processed by CBM3D and DnCNN. After applying FE, the compared algorithms failed to perform color-channel registration on the moving white billiard ball and introduced severe artifacts on the billiard table. Since the white billiard ball had few details and features, FE led to poor results. Although the proposed DKPNN also presented some artifacts, its performance was superior to the compared algorithms.

Hence, the results using the generated test set data showed that, compared with other denoising methods, the proposed DKPNN performed better in denoising and color-channel motion artifact removal.

### 4.2. LCTF-ICMOS Noisy Data

LCTF-ICMOS noisy data under four different environmental illuminant levels using a D65 illuminant were captured using the experimental system setup described in Section 4. The four illuminant levels were 5 × 10^−2^ lx, 1 × 10^−2^ lx, 5 × 10^−3^ lx, and 1 × 10^−3^ lx. The captured scenes included static targets and motion targets. Static targets were color checkers, books, dolls, iron boxes, glass mosaic decorative mats, tank models, etc. Motion targets were two moving toy birds. LCTF-ICMOS was run at 10 fps under different illuminant levels.

Since it is difficult to obtain the ground truth images for real noisy data, no-reference image quality assessment (NRIQA) metrics were introduced to evaluate the denoising results, namely, the roughness ρ [47] and the root-mean-square contrast (RMSC) [48]. The equation describing ρ is as follows:(15)$$\rho (I)=\frac{{\left\|h_{1}⊗I\right\|}_{1}+{\left\|h_{2}⊗I\right\|}_{1}}{{\left\|I\right\|}_{1}},$$
where *I* denotes the input image, *h*_1_ denotes the horizontal filter template [1, −1], *h*_2_ denotes the vertical filter template [1, −1] ^T^, and ${\left\|\cdot \right\|}_{1}$ denotes the L_1_ norm. Generally, a larger value of *ρ* indicates a rougher image, i.e., a higher image noise level.

The equation describing RMSC is as follows:(16)$$\mathrm{RMSC}=\sqrt{\frac{1}{MN}\sum_{j=1}^{N}\sum_{i=1}^{M}{\left(I(i,j)-mean(I)\right)}^{2}},$$
where *i* and *j* denote the row and column indices of the pixel, respectively; *M* and *N* denote the numbers of image rows and columns, respectively; and mean () denotes the mean value calculation of image pixels. A smaller value of RMSC indicates a lower image noise level.

The two NRIQA metrics were calculated after converting color images to grayscale images. The denoised results using real noisy data under different illuminant levels are shown in Figure 11. The evaluation metrics for each algorithm are shown in Table 4.

Since the NRIQAs could not effectively distinguish image details from noise during the assessment, we adopted the strategy of manually selecting flat areas in the image, such as the colored checker patches and the gray background panels, so that the NRIQAs could better evaluate the results. The selected flat areas are shown in the green boxes in Figure 12.

As the environmental illuminant levels decreased, the performance of each denoising algorithm on LCTF-ICMOS real noisy data gradually worsened due to the sharp decrease in the image signal-to-noise ratio. As shown in Figure 11, the image characteristics of LCTF-ICMOS real noisy data are different from the Adobe240 dataset used for network training, which could worsen the performance of proposed DKPNN on both image denoising and motion artifact removal to some extent.

In terms of denoising, the adopted training image noise model is the PGN model as shown in Equation (4). The PGN model can simulate the imaging noise of CMOS/CCD sensors well, however, it is different from the image noise characteristics of LCTF-ICMOS. As shown in Figure 11a, bright spot noise can be observed in LCTF-ICMOS images. The bright spot noise is introduced by the microchannel plate in the image intensifier. The difference in noise model worsened the performance of each denoising algorithms in varying degrees. DnCNN, fDnCNN, and FFDNet performed denoising through learning the mapping between noisy and clear image pairs. Using this training strategy, when the noise models or image characteristics of training and test data are different, the performance would be worsened significantly. The denoising strategy of proposed DKPNN is essentially to aggregate similar pixels in the input image and perform weighted average, which limits the output space through directly manipulating the input pixels. This strategy weakens the impact of noise model differences. As shown in Figure 11f, the proposed DKPNN performed better than the compared algorithms in the flat areas. In order to solve the problems mentioned above, we would further study the image noise model of LCTF-ICMOS and train the DKPNN with the new noise model in future works.

In terms of color-channel motion artifact removal, the proposed DKPNN does not produce as satisfactory results on the ICMOS data as on the generated data. Firstly, the differences between ICMOS data and training Adobe240 data caused the decrease of offsets prediction accuracy of the offset-prediction subnet, thereby affecting the performance of motion artifact removal. Secondly, the difference in noise model also affects the offsets prediction accuracy. Similarly, it also affects the matching accuracy of FE algorithm. Finally, the toy bird moving targets used in the experiment has few detailed features in the ICMOS images. Similar to the billiard ball target in scene 4 of Figure 10, it is challenging to remove motion artifacts from images of such objects. However, the proposed DKPNN still performed better than the compared algorithms in maintaining the edges and details of the moving toy birds. Moreover, FE would introduce severe artifacts at the edges of input images. In order to solve the problems mentioned above, we would capture images of different still scenes using the LCTF-ICMOS setup, and generate images with motion artifacts using the method described in Section 2.2 for network training in future works.

The offset-prediction subnet of the proposed DKPNN adopted the trilinear interpolation method when calculating resampled pixels. The interpolation in the direction of the color channel caused a decrease in color saturation. As shown in scene 3 of Figure 10h, the human hands are grayish compared with the ground truth color image. Though the weight-prediction subnet would give small weights to resampled pixels with large color bias, it is still inevitable that color bias would be introduced especially in areas with severe motion artifacts. To solve this problem, designing a new loss function with a color-bias term or introducing additional color restoration modules utilizing local characteristics of input image and the denoised image can be considered.

Therefore, the proposed DKPNN performed better than the compared algorithms on both generated data and real noisy data. It can be further used to resolve the challenges related to the noised face recognition problem [49].

## 5. Conclusions

Image intensifiers have been used internationally as advanced military night-vision devices for many years due to their superb imaging performance in low-light-level conditions. ICMOS/ICCD was developed by coupling the phosphor screen of the image intensifier to the CMOS/CCD sensor, thereby digitizing the output of the image intensifier. An LCTF-based color imaging ICMOS experimental system was set up in this article; however, due to the time-division color imaging scheme, if there were moving targets in the scene, color-channel motion artifacts were introduced in the final image output. To solve this problem, DKPNN was proposed for joint denoising and motion artifact removal, while a data generation method for color-channel motion artifacts was also proposed to train the proposed DKPNN. The results show that, compared with other denoising algorithms, the proposed DKPNN performed better on both generated noisy data and LCTF-ICMOS real noisy data, whereby it could better maintain the image details. Since the trilinear interpolation method was adopted for pixel resampling in the proposed DKPNN, it lost color performance while applying motion artifacts removal. The size of the LCTF-ICMOS imaging system is relatively large due to the coupling strategy of the relay lens, while the incident energy is partially reduced. Therefore, in future work, corresponding color restoration methods should be studied. The training strategy could also be improved by trying more advanced data augmentation methods based on generative adversarial networks [50]. The phosphor screen in the image intensifier can be removed and a more effective coupling method can be used to further improve the signal-to-noise ratio and greatly reduce the size of the imaging system.

## Figures and Tables

**Figure 1 sensors-21-03891-f001:**
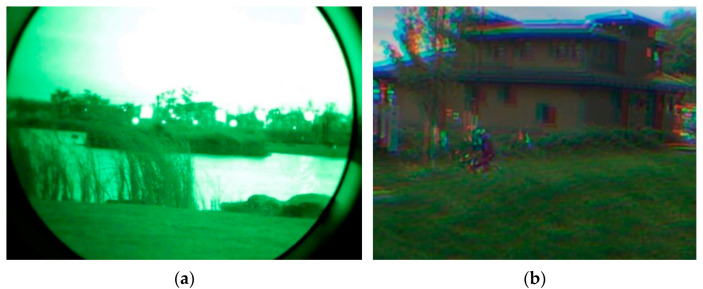
Imaging from image intensifiers and color-channel motion artifacts introduced by time-division color imaging: (**a**) typical monochromatic image from an image intensifier; (**b**) color-channel motion artifacts.

**Figure 2 sensors-21-03891-f002:**
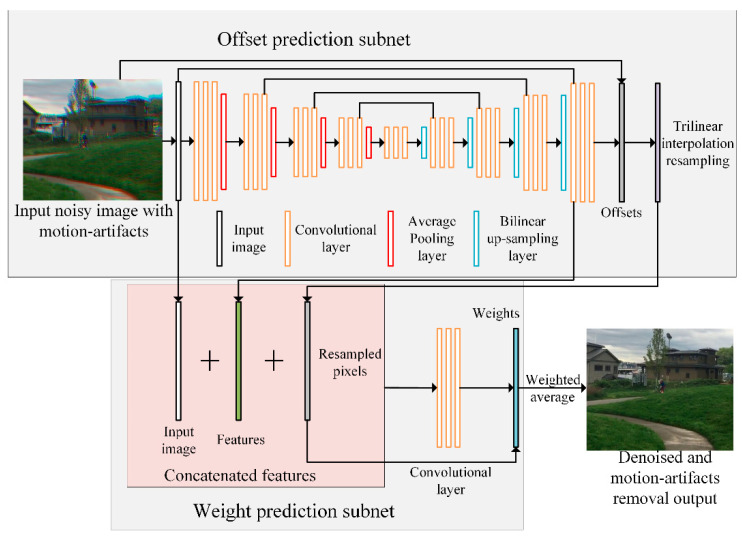
The pipeline and network structure of the proposed DKPNN.

**Figure 3 sensors-21-03891-f003:**
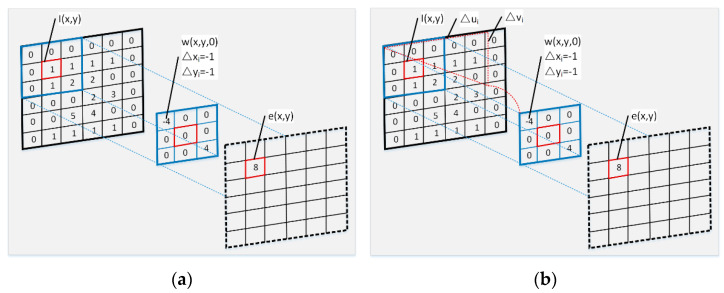
The filtering processes of a typical rectangular filter kernel and a deformable filter kernel: (**a**) rectangular filter kernel; (**b**) deformable filter kernel.

**Figure 4 sensors-21-03891-f004:**
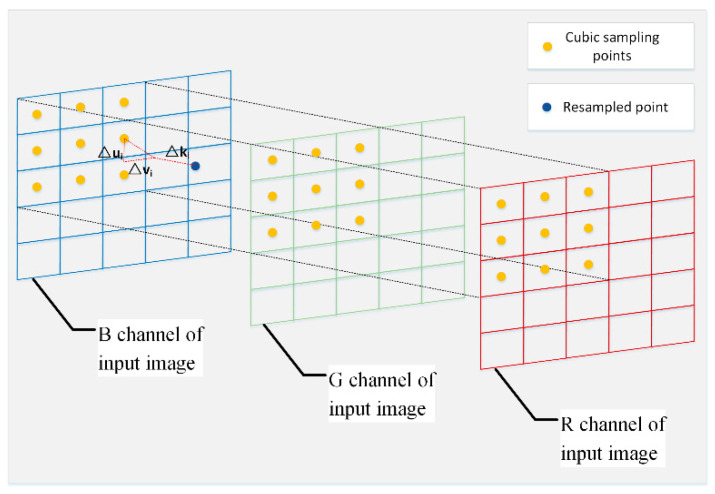
The learning procedure of three-dimensional deformable sampling locations.

**Figure 5 sensors-21-03891-f005:**
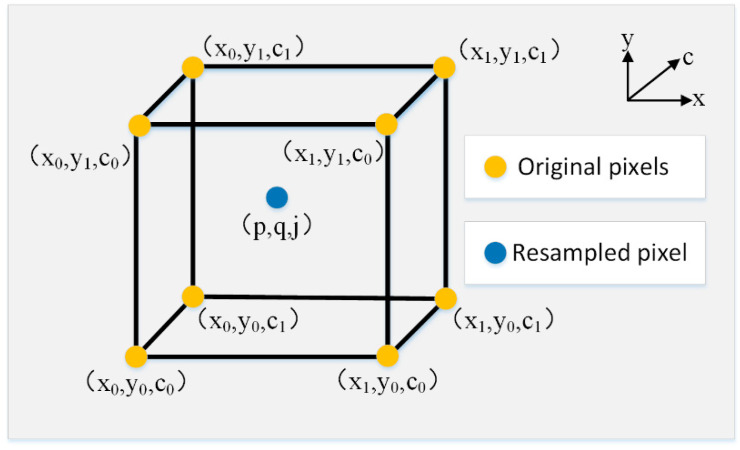
The resampling process based on trilinear interpolation.

**Figure 6 sensors-21-03891-f006:**
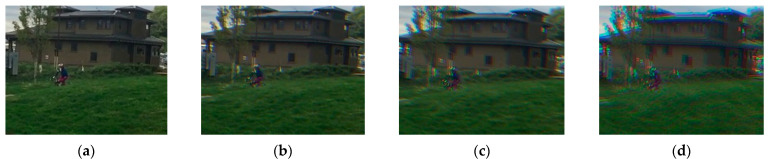
The generated images with different motion severity: (**a**) ground truth; (**b**) small motion; (**c**) medium motion; (**d**) big motion.

**Figure 7 sensors-21-03891-f007:**
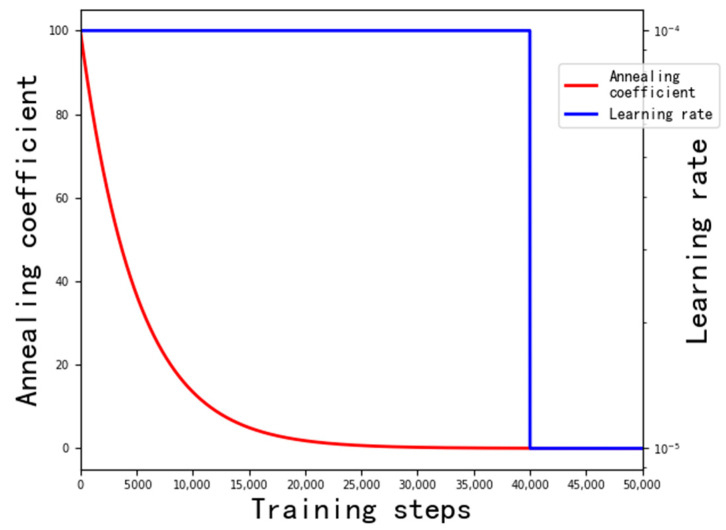
Relationship between the number of training steps and the coefficient of the annealing term and the learning rate.

**Figure 8 sensors-21-03891-f008:**
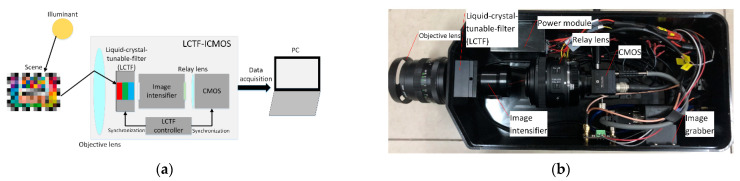
LCTF-ICMOS color imaging system: (**a**) imaging process of the system; (**b**) system setup with shell opened.

**Figure 9 sensors-21-03891-f009:**
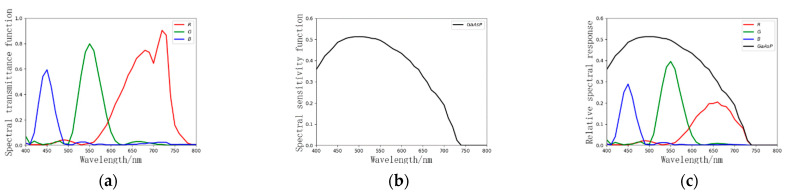
Spectral sensitivity function of LCTF-ICMOS: (**a**) spectral transmittance function of LCTF; (**b**) spectral sensitivity function of image intensifier; (**c**) relative response function of LCTF and image intensifier.

**Figure 10 sensors-21-03891-f010:**
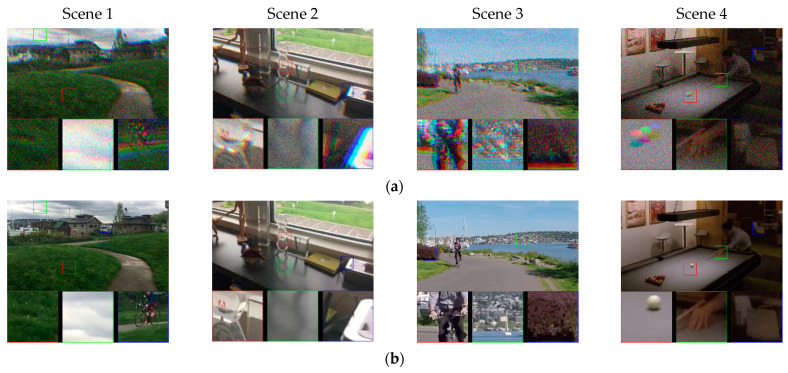
Denoising results on generated test set data: (**a**) input noisy image with color-channel motion artifacts; (**b**) ground truth; (**c**) CBM3D; (**d**) CBM3D + FE; (**e**) DnCNN + FE; (**f**) fDnCNN + FE; (**g**) FFDNet + FE; (**h**) proposed DKPNN.

**Figure 11 sensors-21-03891-f011:**
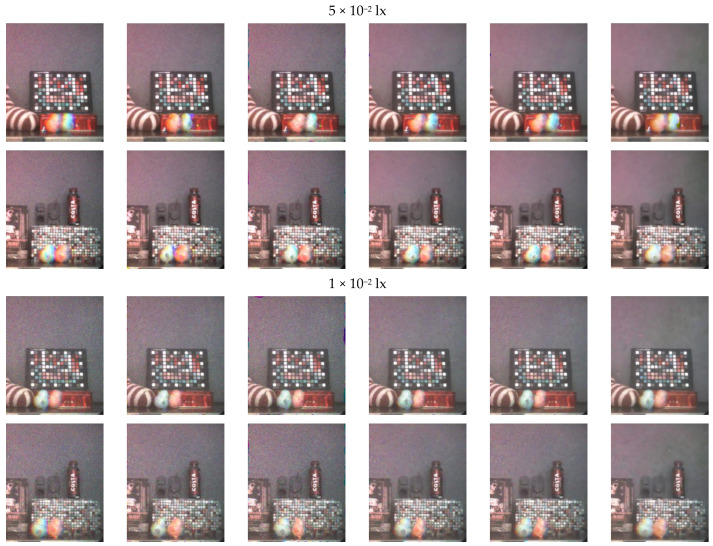
Results using real noisy data under different illuminant levels, with scene 1 on the top row and scene 2 on the bottom row: (**a**) noisy input; (**b**) CBM3D + FE; (**c**) DnCNN + FE; (**d**) fDnCNN + FE; (**e**) FFDNet + FE; (**f**) proposed DKPNN.

**Figure 12 sensors-21-03891-f012:**
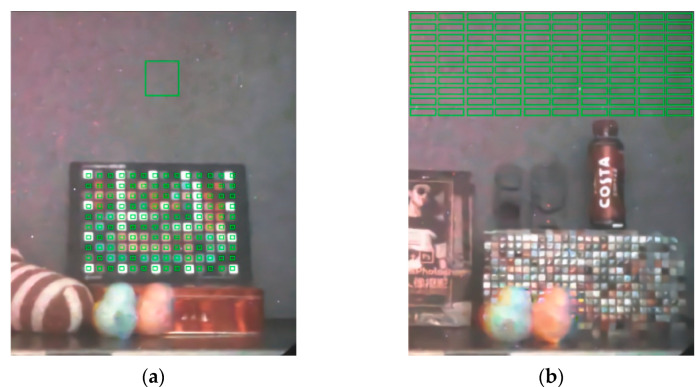
The selected flat areas for NRIQA calculation: (**a**) scene 1; (**b**) scene 2.

**Table 1 sensors-21-03891-t001:** The number of channels in each convolutional layer of the offset-prediction subnet.

Convolutional Layers	L1–3	L4–6	L7–9	L10–12	L13–15	L16–18	L19–21	L22–24	L25–26	L27
Number of channels	64	128	256	512	512	512	256	128	128	*n* × 3 × 3

**Table 2 sensors-21-03891-t002:** The number of channels in each convolutional layer of the weight prediction subnet.

Convolutional Layers	L1	L2	L3
Number of channels	64	64	*n* × 3

**Table 3 sensors-21-03891-t003:** The evaluation metrics on the test set data for each algorithm.

Noise Level	Metrics	BM3D	DnCNN	fDnCNN	FFDNet	BM3D + FE	DnCNN + FE	fDnCNN + FE	FFDNet + FE	DKPNN
$\sigma_{p}=2\times 10^{-3}$ $\sigma_{g}=2\times 10^{-2}$	PSNR	22.68	23.02	22.88	22.84	25.06	26.50	26.58	26.51	29.33
	SSIM	0.721	0.732	0.726	0.725	0.818	0.831	0.834	0.832	0.887
$\sigma_{p}=4\times 10^{-3}$ $\sigma_{g}=3\times 10^{-2}$	PSNR	21.56	22.33	22.01	22.11	24.34	25.78	25.80	25.74	28.68
	SSIM	0.712	0.722	0.715	0.716	0.806	0.820	0.826	0.823	0.878

**Table 4 sensors-21-03891-t004:** Evaluation metrics for each algorithm under different illuminant levels.

Metrics	Scene	Illuminant Levels (lx)	Algorithms					
			Noisy Input	BM3D	DnCNN	fDnCNN	FFDNet	Proposed
ρ	1	5 × 10^−2^	0.0412	0.0117	0.0248	0.0079	0.0079	0.0059
		1 × 10^−2^	0.0577	0.0128	0.0286	0.0103	0.0103	0.0081
		5 × 10^−3^	0.0719	0.0144	0.0291	0.0111	0.0108	0.01
		1 × 10^−3^	0.0778	0.0148	0.0302	0.0114	0.0113	0.0108
	2	5 × 10^−2^	0.0436	0.0099	0.0269	0.0059	0.0058	0.0042
		1 × 10^−2^	0.0601	0.0153	0.0273	0.0112	0.0113	0.0086
		5 × 10^−3^	0.0721	0.0154	0.0306	0.013	0.0131	0.0111
		1 × 10^−3^	0.0807	0.0165	0.0308	0.0133	0.0132	0.0125
RMSC	1	5 × 10^−2^	8.1287	5.6751	7.299	5.6361	5.7168	5.4267
		1 × 10^−2^	10.8123	6.5564	8.972	6.0954	6.2002	5.4395
		5 × 10^−3^	14.2621	8.7327	11.3297	7.9825	8.206	6.9467
		1 × 10^−3^	17.2388	11.8769	14.0814	10.9492	11.3108	9.9249
	2	5 × 10^−2^	8.8464	3.7097	7.0186	2.3180	2.2060	1.9741
		1 × 10^−2^	11.9845	5.5569	9.2652	3.7849	3.7955	2.1758
		5 × 10^−3^	13.5307	6.2826	10.0309	4.5214	4.5867	2.5442
		1 × 10^−3^	15.3455	7.8517	11.7887	5.8253	5.7143	3.5552

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
