# Peer review of "Denoising and Motion Artifact Removal Using Deformable Kernel Prediction Neural Network for Color-Intensified CMOS"

_sensors, 2021, doi:10.3390/s21113891_

Round 1

Reviewer 1 Report

The proposed approach for handling both motion artefacts and noise for LCTF-ICMOS imaging system using spatially varying sampling grids and weights is interesting and seems to be able to produce better results than the considered approaches.

The paper is in general well written. In my opinion, some further explanations are necessary for an easier understanding of its content:

  • fig. 2 shows in the pipeline a bilinear interpolation whereas fig.5 refers to a trilinear interpolation. This should be explained.
  • with respect to handcrafted features or mathematically derived denoising methods in CNN-based denoising one can seldom see  handcrafted architectures. From this point of view, the architecture in fig.2 should be better explained, the choice of the hardcoded values from table 1 and table 2 should also be justified.
  • the authors argue that the approach that they extend in their method (refs. 32 and 33 ) cannot be used directly to handle the type of images the LCTF-ICMOS sensor produces). This should be better explained in the paper.
  • data augmentation is done using random horizontal and vertical flips. There are more modern approaches such as generative adversarial networks that could be considered. 
  • for me, in the current form, it is not clear noise was handled during training (how many values were considered for the noise parameters in eq. 4) and how does the network perform when a noisy input with different noise parameters is to be restored.

Reviewer 2 Report

The manuscript by Han et al. presents a denoising and motion artefact removal method using Deformable Kernel Prediction Neural Network for color-intensified CMOS scenario. Overall, the manuscript is well written and the content is complete. However, the following concerns have to be addressed before its publication in Sensors.

1. I am not sure why the literature review on image denoising is put in section 2 "Related Work" instead of integrated into "introduction". Besides, this review on previous work is very scattered without pointing out their deficiencies in the color-intensified CMOS scenario, resulting in weak presence of the significance of this work. The authors should better address the "so-what?" question in the introduction part.

2. All the figures starting from Fig.2 are very blurred. It is very challenging to read the figure contents. 

3. The training of the proposed DKPNN is based on Adobe240 database which is dramatically different from the color-intensified CMOS imagery. The author should discuss how this implementation will impact the performance of the trained network in terms of denoising and motion correction.

4. In Section 5 Results and Discussion, the results from the generated noisy data is satisfying while the improvement is not obvious from the LCTF-ICMOS data despite that the proposed method outperforms other methods in the metrics of roughness and RMSC. Is this phenomenon related to the way how the network is trained? A better discussion on this part is expected.

5. As you authors pointed out in the conclusions, color restoration is challenging yet important for LCTF-ICMOS applications where color information is thought to be crucial. Please add some discussion on the color performance in section 5.2.
